# Vitamin D as A Protector of Arterial Health: Potential Role in Peripheral Arterial Disease Formation

**DOI:** 10.3390/ijms20194907

**Published:** 2019-10-03

**Authors:** Smriti Murali Krishna

**Affiliations:** College of Medicine and Dentistry, James Cook University, Townsville, QLD 4811, Australia; smriti.krishna@jcu.edu.au; Tel.: +61-400-463-101

**Keywords:** vitamin D, 25(OH)D, peripheral arterial disease, peripheral arterial occlusive disease, abdominal aortic aneurysm, epigenetics

## Abstract

Atherosclerotic occlusive diseases and aneurysms that affect large and medium-sized arteries outside the cardiac and cerebral circulation are collectively known as peripheral arterial disease (PAD). With a rise in the rate of aging population worldwide, the number of people diagnosed with PAD is rapidly increasing. The micronutrient vitamin D is an important steroid hormone that acts on many crucial cellular mechanisms. Experimental studies suggest that optimal levels of vitamin D have beneficial effects on the heart and blood vessels; however, high vitamin D concentrations have been implicated in promoting vascular calcification and arterial stiffness. Observations from various clinical studies shows that deficiency of vitamin D has been associated with a greater risk of PAD. Epidemiological studies have often reported an inverse relation between circulating vitamin D status measured in terms of 25-hydroxivitamin D [25(OH)D] levels and increased cardiovascular disease risk; however, randomized controlled trials did not show a consistent positive effect of vitamin D supplementation on cardiovascular disease risk or events. Even though PAD shares all the major risk factors with cardiovascular diseases, the effect of vitamin D deficiency in PAD is not clear. Current evidence suggests a strong role of vitamin D in promoting genomic and epigenomic changes. This review summarises the current literature that supports the notion that vitamin D deficiency may promote PAD formation. A better understanding of underlying pathological mechanisms will open up new therapeutic possibilities which is the main unmet need in PAD management. Furthermore, epigenetic evidence shows that a more holistic approach towards PAD prevention that incorporates a healthy lifestyle, adequate exercise and optimal nutrition may be more effective in protecting the genome and maintaining a healthy vasculature.

## 1. Introduction

Vitamin D is an important micronutrient that is essential for optimal health throughout life. Recent research suggest that vitamin D may possess a wide range of biological effects beyond its classically recognized function in bone and mineral homeostasis [1]. Indeed, deficiency of vitamin D has been associated with increased prevalence of multiple diseases including osteoporosis, a number of autoimmune diseases, many different cancers and conditions such as hypertension and cardiovascular diseases (CVDs) [2].

Vitamin D, with D3 (cholecalciferol) and D2 (ergocalciferol) being the most important for human health, is obtained through two main sources, with UV light-dependent synthesis of cholecalciferol from cholesterol in the skin being the predominant source of optimal vitamin D levels (Sunlight is the major natural source of vitamin D). To a lesser degree, vitamin D may also be obtained through diet (dairy products, eggs or fatty fish being examples) or supplements as vitamin D3 or vitamin D2 [2] (Figure 1). As vitamin D is naturally found in a limited range of foods, fortified foods often provide supplementation required for maintaining optimal health.

Deficiency of vitamin D has been previously reported to be common worldwide [3,4] with approximately 50% of the global population exhibiting lower circulating levels of vitamin D [5,6,7]. A systematic review by Hilger J, et al. (assessing 195 studies published in 1990–2011 in 44 countries, with more than 168,000 participants) highlights that people residing in North America had higher circulating vitamin D levels compared to Europe, Middle East or Africa, furthermore a difference in circulating levels were reported between the European countries [3]. Given that vitamin D synthesis is mainly through sunlight, factors such as the skin tone, availability of sunlight, etc. may account for these differences in vitamin D levels.

In recent research, inadequate vitamin D status has been linked to non-skeletal major disease such as CVD [8]. Of particular interest to this review, is the emerging concept of the association of vitamin D in cardiovascular health and the contribution of its deficiency towards the development and progression of peripheral arterial disease (PAD). While patients with PAD often have a sedentary lifestyle with limited exposure to sunlight. Age may also be a factor, with the prevalence of PAD being greater in the elderly. Interestingly, the capacity of UV-mediated vitamin D synthesis was previously reported to be reduced in aged skin, with an associated reduced expression of the vitamin D receptor in aged human muscle [9]. As such, dietary supplementation of vitamin D for such patients may be beneficial towards reducing the risk of CVD/PAD, as well as improving patient outcomes. Hence, it is of interest to know the pattern of PAD prevalence and the link with vitamin D status if any, to inform public health policy development to reduce risk for potential health consequences of an inadequate vitamin D status. Furthermore, it is vital to understand whether the association of vitamin D and PAD is causative or is reverse causality. However, the presence of receptors for vitamin D in the vascular wall suggests that this micronutrient might play a role in the pathogenesis of arterial diseases [10]. In this review, the role of vitamin D in maintaining arterial health is assessed and the effect of vitamin deficiency on the genome and beyond is explored, with special relevance to PAD. 

## 2. Peripheral Arterial Disease (PAD)

Atherosclerotic occlusive conditions and aortic aneurysms that affect large and medium-sized arteries outside the cardiac and cerebral circulation are collectively known as PAD [11]. Specifically, PAD comprises of the pathological changes occurring in the abdominal aorta, iliac and lower-extremity arteries. Peripheral arterial occlusive disease (PAOD) or lower extremity arterial disease is an athero-thrombotic disease leading to stenosis or occlusion of the leg arteries, and eventual death of tissue due to continued ischemia. Abdominal aortic aneurysms (AAA) are degenerative diseases characterised by extensive inflammation and extracellular matrix (ECM) remodelling and eventual destruction of medial layer of aorta resulting in its rupture. Patho-physiologically, both PAOD and AAA are multifactorial conditions with multi-phase complications. A number of risk factors are currently known, several of these are known to cause alterations to the genome and epigenome. It is evident that there is an urgent need of new and better therapies and to investigate other factors that are associated with the disease’s presence and progression. A detailed description of the pathology of PAD is beyond the scope of this review and readers are advised to refer to the previous excellent reviews on this topic [12,13]. 

### 2.1. Vitamin D Status and PAOD Formation

PAOD occurs due to an impediment of the blood flow to the lower extremities and manifests as intermittent claudication (IC) or leg pain during walking and critical limb ischemia (CLI) where there is tissue loss [14]. In primary care, diagnosis of PAOD is performed by measuring the ankle-brachial pressure index (ABPI) with a score <0.9. More than 200 million globally has been estimated to be affected by PAOD [15,16]. Given the trend in the aging population, the prevalence of PAD and lower limb PAOD has been projected to rise in forthcoming years [16,17,18] and is expected that the PAOD prevalence will double by 2040 [19]. A recent meta-analysis highlighted that PAOD poses as a significant global threat in both high-income and low-to-middle income countries, suggesting that the condition is prevalent worldwide [16]. Even in developed countries, PAOD continues to be underreported and undermanaged [20]. PAOD patients are at approximately three times higher risk of cardiovascular and all-cause mortality despite the currently available effective medical therapy [21]. This results in poor outcome for the patient, with chronic claudication pain, ischemic ulcerations, repeated hospitalizations, revascularizations, limb necrosis and eventual limb loss. As a consequence, these problems result in an overall poor quality of life and a high rate of depression among PAOD patients [22]. Hence, better and effective treatments are urgently required to reduce the high risk of lower limb and cardiovascular events in this population.

Current studies strongly suggest a correlation between reduced circulating vitamin D levels and presence of PAOD [23,24,25,26,27,28,29,30,31], although some epidemiological studies have shown no correlation [32,33,34]. Table 1 shows cross-sectional studies investigating vitamin D levels in circulation in PAOD patients. It should be noted that the threshold for low vitamin D or hypovitaminosis defined by 25(OH)D levels were not similar in these studies, with low vitamin D levels defined in the range from <17.8 ng/mL to <30 ng/mL. This could be a reason for the difference in the correlation shown between the studies. Furthermore, the geographical location of the patient population tested varied between the studies, leading to other confounding factors, which could potentially lead to the change in vitamin D levels.

Data from the *National Health and Nutrition Examination Survey* (NHANES, 4864 participants, 1999–2002) shows that the patients with PAOD had lower plasma levels of vitamin D [35]. Reports from the Atherosclerosis Risk In Communities (ARIC) study shows that lower levels of 25(OH)D was associated with 30% increased risk of PAOD in participants regardless of their ethnicity [23]. Moreover, in another cross-sectional analysis (NHANES, 2001–2004), an 80% higher prevalence of PAOD was reported in participants in the lowest quartile of 25(OH)D compared to the participants in the highest quartile [28]. In a recently performed a meta-analysis on six observational studies, we showed that PAOD patients (*n* = 1217) had lower 25(OH)D compared to non-PAOD participants (*n* = 5201) [36]. Interestingly, the study also demonstrated that patients with advanced PAOD symptoms of CLI had even lower circulating levels of 25(OH)D when compared with PAOD patients that had IC symptoms [36]. This is corroborated by the observation that patients with PAOD shows progressive functional decline [10] and low vitamin D status was associated with lower ABPI and a faster decline of functional performance [37]. 

As indicated by several studies in humans and in animals, aging results in a decline in the ability of the kidney to synthesize 1,25(OH)_2_D_3_ [38]. Furthermore, PAD patients experience severe debilitating lower limb symptoms which includes, IC, rest-pain, arterial ulcers and gangrene, which severely limit their physical activity [39]. The reduced ability of the kidney to synthesis 1,25(OH)_2_D_3_ and probably the limited exposure to sun which results in the lack of efficient cutaneous production could result in the reduced vitamin D levels seen in the PAD patients. Furthermore, as seen in the CVDs, the reduced vitamin D levels reported in various studies could also be due to different confounding factors including environment, age, sex, socio-economic and nutritional status. 

### 2.2. Vitamin D Status and AAA Formation

Abdominal aortic aneurysm (AAA) is pathologically characterised by progressive degeneration of the arterial wall structure by chronic inflammation and ECM remodelling which leads to irreversible dilatation and eventual rupture resulting in death [13,40]. AAA affects 4–8% of men and 1.5% of women above the age of 60 years and usually remains asymptomatic until rupture [41]. AAA rupture is a medical emergency with a high mortality rate (80–90%) and majority of the deaths occurring before reaching the hospital. Approximately 20 million people are estimated to have AAA [18,42] and given the trend in the aging population, the prevalence of AAA has been projected to rise in forthcoming years [16,17,18].

An interesting population-based cohort study from northern Sweden, showed seasonal hypovitaminosis in the population [43]. A higher prevalence of AAA was reported in the adults and the reported incidence of AAA repair was also higher in the northern region where sunlight is scarce [43,44] showing a possible correlation between vitamin D status and AAA formation. Deficiency of vitamin D was shown to be an independent risk factor associated with thoracic aortic dilatation [45] and an inverse relationship between vitamin D levels and AAA development has also been established [46,47,48]. In an observational study of 4233 community-dwelling men (age range 70–88 years), who participated in a randomised controlled trial of screening for AAA, low vitamin D status in older men was associated with the presence of larger AAA [46]. A graded inverse relationship was observed between circulating 25(OH)D concentrations and AAA diameter. A recent pooled analysis of three case-control studies demonstrated significantly lower circulating 25(OH)D levels in patients with AAA than subjects without AAA [49]. A study utilised redox proteomic approach to investigate total and specific protein carbonylation and protein-bound 4-hydroxy-2-nonenal (HNE) in the serum of AAA patients compared with age-matched controls [50]. The results show specific carbonylation of three serum proteins, which included vitamin D-binding protein (DBP), serum retinol-binding protein and fibrinogen α-chain HNE, suggesting the importance of adequate vitamin D levels in AAA formation. A recent clinical evaluation study on small number of AAA patients (*n* = 11) showed that selective activation of vitamin D receptor (VDR) by paricalcitol (1 μg daily, 24–weeks before open AAA repair) interferes with calcineurin-mediated inflammation in AAA [51]. Paricalcitol is a VDR agonist and the mechanism of action was shown to be anti-inflammatory action by reduction in T cell activation. To date, there are no large-scale clinical trials assessing the link between vitamin D supplementation and AAA, therefore clear evidence of a treatment effect is lacking. 

Contradictory studies also exist showing that confounding factors could play a role in determining the association between vitamin D status using currently available biomarkers (vitamin D metabolic markers) and AAA presence. For example, a recent prospective epidemiologic study examined the association between serum 25(OH)D concentrations and incidence of AAA in the ARIC cohort [52]. However, in this large prospective cohort (12,770 participants), there was no association between 25(OH)D concentrations and risk of AAA presence. The DBP is the major plasma carrier protein of vitamin D and used as a biomarker of vitamin D metabolism. However, the presence of DBP and its use as a reliable biomarker in AAA is contradictory. A previous proteomic study showed that DBP levels positively correlated with the size and expansion of AAA [53], however another proteomic analysis of plasma proteome of patients undergoing AAA repair showed a negative correlation between the DBP and AAA presence [41]. It should also be noted that in most of these studies, the biomarkers are measured as a one-off measurement and may not be possibly coinciding with the natural history of AAA [52]. Furthermore, recently it was highlighted that unbound free-25(OH)D might be a better reliable biomarker of vitamin D status [54]. These observations suggest that further studies are required to understand which vitamin D metabolite is optimal in ascertaining vitamin D status [54].

Vitamin D_3_ seems to act on a range of cellular and molecular mechanisms important to the pathogenesis of AAA [55] (Figure 2). Evidence from preclinical studies shows that calcitriol treatment significantly attenuates dissecting AAA formation [56]. Oral treatment with calcitriol reduced angiotensin II (AngII)-induced dissecting AAA formation in apolipoprotein E deficient (*ApoE^-/-^*) mice with marked increase in VDR-retinoid X receptor (RXR)-α interaction in the aortas of calcitriol-treated mice. Calcitriol-co-treated mice also exhibited reduced macrophage infiltration, matrix metalloproteinase (MMP) and chemokine expression in the suprarenal aortic walls, which seems to be mediated specifically through activation of VDR–RXR-α interactions [56]. Increasing evidence suggests that vitamin D3 is a potent endogenous regulator of the renin–angiotensin-aldosterone-system (RAAS) [57,58,59]. Accordingly, VDR-deficient mice display high renin and AngII levels and develop hypertension [57]. In addition to regulating the components involved in synthesizing AngII, calcitriol also alters the expression of AngII, AT1 receptors in several target tissues [60,61]. *In vitro* studies also showed that VDR activation by calcitriol in human endothelial cells (ECs) inhibited AngII-induced leukocyte-EC interactions, morphogenesis, and production of endothelial pro-inflammatory and angiogenic chemokines through VDR-RXR interactions, and knockdown of VDR or RXR abolished the inhibitory effects of calcitriol [56].

## 3. Vitamin D Status and Mechanisms Relevant in PAD Formation

Vitamin D has a range of biological functions and affect many crucial mechanisms and cell types that maintain a healthy vasculature (Figure 3). Insufficient levels of vitamin D may affect the vasculature via the classic mechanism of promoting changes in the calcium-phosphate metabolism, or via effects on the RAAS [57]. The RAAS is an important endocrine system which controls peripheral vascular resistance, vascular tone and volume homeostasis and the major vascular protective effect of vitamin D is attributed to also its ability to upregulate nitric oxide (NO) which regulates RAAS [62,63]. 

### 3.1. Aortic Cells and Dysfunction

Vitamin D has been proposed to be involved in the regulation of many cell types involved in PAD formation, such as infiltrating/circulating immune cells, as well as residential vascular cells, such as ECs and vascular smooth muscle cells (VSMCs) [64,65] (Figure 3). Indeed, the presence of VDR on VSMCs and ECs suggests that Vitamin D may have a direct role in the regulation of vascular function [66]. In vitro, 1,25(OH)2D_3_ has been shown to be a potent modulator of the growth of VSMC [67], imparts an inflammatory effect upon ECs [68] playing a role in accelerated proliferation increased carotid intima-media thickness (IMT). Active form of vitamin D, calcitriol has previously shown to promote proliferation in VSMCs [69] and promote upregulation of endothelin and NO in cultured ECs [70]. Both EC and VSMCs are also able to locally produce calcitriol from calcidiol [71]. Both cell types are shown to downregulate the production of atherogenic molecules when exposed to vitamin D [72], suggesting that optimal vitamin D status is crucial for the functioning of these cells. 

As the major vascular protective role of vitamin D is NO upregulation which directly affects the ECs, it can be assumed that ECs will be the first responders of any alteration in vitamin D status. Vitamin D has shown to reduce endoplasmic reticulum stress and oxidative stress in EC thereby reducing atherosclerosis [73]. Attenuated NO production in rat aortic ECs was restored by 12 h of calcitrol treatment [61] and the NO production was shown to occur via the interaction with VDR [74] suggesting a direct role of vitamin D in protecting the vascular endothelium from dysfunction. *In vivo* testing of calcitrol in rats showed an increase in the levels of endothelin-1 in plasma and tissue NO production as well as an upregulation of endothelial NO synthase enzyme (eNOS) content in aorta. Mice with deficiency of VDR (*Vdr^-/-^*) have increased levels of renin and consequently of AngII, resulting in hypertension and cardiac hypertrophy, lipid metabolism, cellular stress response, and changes in the vascular function [57]. The lack of *Vdr* signalling results in chronically lower NO bioavailability due to reduced expression of eNOS enzyme, and these effects are independent of changes in the RAAS [75]. *Vdr* null mice display a pro-thrombotic state that was associated with a decrease in antithrombin and thrombomodulin [76]. EC specific deletion of *Vdr* in mice results in endothelial dysfunction evidenced by impaired blood vessel relaxation, an effect that was associated with reduced eNOS expression [77]. Calcitriol has shown to attenuate human glycated albumin-induced IL-6 and IL-8 production in fibroblasts [78], indicating a potential protective effect in AAA where fibroblast plays crucial role in maintaining matrix integrity.

### 3.2. Atherosclerosis

Atherosclerosis is the primary cause of PAD and is characterized by focal lumen-narrowing by atheromatous plaques in the intima of large and medium-sized vessels. Patients with severe atherosclerosis in the coronary arteries have a significantly lower 25(OH)D level compared to patients without significant lesions [79] suggesting a possible similar role in PAD. Vitamin D has been shown to promote the expression of anti-atherogenic monocyte/macrophage subtypes and downregulate the production of various molecules involved in atherogenesis by the ECs and VSMCs [72] suggesting a role of adequate levels of vitamin D in protecting vasculature from the formation and progression of atherosclerosis [62].

Calcitriol has been shown to prevent atherosclerotic plaque destabilization in diabetic individuals by inhibiting the transformation of macrophages to foam cells [80] and the suppressive effect on MMP activity [81]. It should be noted that in macrophages from diabetic individuals, the activation of VDR signalling directly leads to a reduction in the uptake of modified low-density lipoprotein cholesterol, thereby preventing foam cell formation. Vitamin-deficient participants had larger aortic IMT, a predictive marker for subclinical atherosclerosis, compared to those with normal vitamin D levels [82]. Vitamin D deficiency was highly prevalent in patients with occlusive and aneurysmatic arterial disease, independent of traditional CVD risk factors, and showed a strong association with the severity of the arterial disease and atherosclerotic markers such as carotid artery IMT, ABPI and high-sensitive CRP [10]. Deficiency of vitamin D has been identified as a potential risk factor for PAD, if 25(OH)D_3_ levels decrease below 10 ng/mL (OR, 1.35; 95% CI, 1.15–1.59) [28,83]. Interestingly, low circulating 25(OH)D has also been associated with accumulation of unfavourable blood lipid profiles [84], another common factor among PAD patients [85].

Vitamin D has been shown to be effective at inhibiting atherosclerosis lesion formation in animal models [62,86,87]. Oral administration of calcitriol decreases atherosclerosis in *ApoE*^−/−^ mice by decreasing macrophage accumulation and inducing T regulatory cells (T_regs_) and immature dendritic cells with tolerogenic functions [86]. Vitamin D supplementation protected hypercholesterolemic Yucatan microswine against atherosclerosis by controlling cholesterol efflux and macrophage polarization [87]. Additionally, 1,25(OH)_2_D_3_ was shown to increase the expression of liver X receptors, ATP-binding membrane cassette transporter A1, and ATP-binding membrane cassette transporter G1 and promoted cholesterol efflux in THP-1 macrophage-derived foam cells. Collectively, the findings appear to reinforce the notion that low vitamin D could potentially contribute to PAD pathogenesis through mechanisms that are directly and indirectly related to thrombo-atherogenesis lending support to the observed athero-occlusive condition in PAD patients.

### 3.3. Inflammation

Inflammatory cells such as monocyte/macrophage plays major role in atherosclerosis formation and vitamin D has been shown to promote the expression of anti-atherogenic monocyte/macrophage subtypes [72]. Vitamin D has an important role in the immune system as it has been shown to have immune-modulating anti-inflammatory activities [88] and its deficiency has been implicated in systemic and vascular inflammation [89,90]. 1,25(OH)_2_D_3_ regulates both innate and adaptive immunity in opposite directions, i.e.; it promotes the innate immune response and inhibits the adaptive immune response [91]. The immunosuppressive effect of 1,25(OH)_2_D_3_ is mediated through a decrease in inflammatory cytokines including interleukin (IL)-2 and interferon (IFN)-γ [92]. Immunomodulatory property of 1,25(OH)_2_D_3_ has also been shown to be mediated through an enhancement of T_regs_ and through the induction of transcription factor Foxp3, which is crucial in the development and function of T_regs_ [1]. 

Clinical studies suggest that low levels of vitamin D_3_ are associated with inflammation and the consequent endothelial dysfunction [93]. An observational study (Irish adults aged >60 years, *n* = 957) showed that low circulating 25(OH)D was associated with increased inflammation, including elevated IL-6 and hs-CRP, paralleled by reduced levels of anti-inflammatory cytokine IL-10 [94]. Low circulating 25(OH)D has previously been associated with blood lipid profiles [84], and macrophages harvested from obese, diabetic and hypertensive patients showed that vitamin D limited foam cell formation and improved high density lipoprotein transport [80] suggesting that the protective effect might be mediated through the action on macrophages.

1,25(OH)_2_D_3_ interferes with the maturation of dendritic cells and maintains the dendritic cells in a semi-mature state and in vivo exposure of mouse dendritic cells to 1,25(OH)_2_D_3_ results in restoring their functional migratory capacity and decrease the proliferation of activated T cells [95]. Administration of calcitriol induces two major immune cells induced in atherosclerotic lesions namely, T_regs_ and immature dendritic cells via the intestinal immune system, resulting in mutually suppressing pathogenic immune processes that play a pivotal role in the progression of atherosclerosis [86]. In *ApoE^-/-^* mice, supplementation of calcitriol was shown to attenuate atherosclerosis formation by increasing T_reg_ subsets and decreasing differentiation of dendritic cells or by activation of VDR [86,96]. Calcitriol has been shown to be anti-atherosclerotic in *ApoE**^-/-^* mice by attenuating macrophage accumulation, promoting functions of T_regs_ and dendritic cells [86] and in swine by promoting cholesterol efflux and macrophage polarization [87]. Calcitriol has been shown to have anti-inflammatory properties and the inhibitory effect was via supressing the production of IL-6 and tumour necrosis factor (TNF)-α [97]. Evidence clearly shows a major anti-inflammatory role of vitamin D, however, understanding of mechanisms by which vitamin D deficiency alters pathological events in PAD remains elusive. 

### 3.4. Arterial Stiffness and Calcification

Vascular calcification is a pathology characteristically observed in advanced age and localised intimal calcification is a feature of atherosclerosis. Extensive medial calcification is a common presenting feature of PAD and is visible even on plain X-ray [98]. Medial arterial calcification is associated with increased pulse pressure and arterial stiffness [99]. Vascular stiffness is an established predictor of cardiovascular morbidity and mortality and deficiency of vitamin D has also been associated with vascular stiffness [90]. An inverse association has been shown between 25(OH)D_3_ and parameters of endothelial dysfunction [93,100] and arterial stiffness [93], suggesting a role for hypovitaminosis in PAD formation. It is interesting to note that arterial stiffness was generally assessed by brachial artery flow-mediated dilation [93], which is a common assessment used to determine the effect of physical activity in PAD patients.

A randomised control trial involving a small number of black youth (*n* = 49), showed that daily supplementation of vitamin D (2000 IU) improved arterial stiffness [101]. A randomised placebo-controlled trial in non-diabetic chronic kidney disease (CKD) patients with low vitamin D levels who were supplemented with oral vitamin D_3_, showed a significant improvement in brachial artery flow-mediated dilation indicating a protective effect of vitamin D on endothelial/vascular function [102]. Aortic stiffness and decreased vessel compliance has been reported in hypertensive patients and a significant antihypertensive effect of vitamin D was reported in hypertensive patients deficient in vitamin D [103]. Supplementation with vitamin D also decreased arterial stiffness as indicated by decreased pulse wave velocity showing the vascular protective effect of optimal vitamin D levels. However, preclinical studies show that caution must be employed when administering vitamin D supplementation, as lower levels of paricalcitol administration was protective against aortic calcification, however at higher dose it induces aortic calcification [104]. 

### 3.5. Vitamin D Status and Angiogenesis

ECs are the primary constituents of new vessel formation which occurs by a process called angiogenesis. A variety of endothelial functions are required for efficient angiogenesis to occur. PAOD is characterised by reduced angiogenesis due to continued ischemia to distal tissues. Cell proliferation is an important step in angiogenesis and it is shown previously that vitamin D_3_ influences cell cycle and proliferation through a vascular endothelial growth factor (VEGF)-mediated pathway [69,105]. In vitro evidence shows a significant effect of vitamin D_3_ on angiogenic potential of endothelial progenitor cells (EPCs) [105]. A significantly higher proliferative rate and increased formation of the whole length of capillary-like structures on matrigel was observed when EPCs were treated with vitamin D3. The increase capillary-formation observed after vitamin D_3_ treatment in EPCs in comparison to controls could be mediated by the increased expression of VEGF that is known to stimulate the proliferation, migration and differentiation of ECs. Vitamin D has been shown to regulate NO production and endothelin-1 in EC in vitro [70], which also stimulates angiogenesis.

The effect of vitamin D on angiogenesis is ambiguous. Preclinical animal model studies show conflicting data on the effect of vitamin D status on angiogenesis. Animal model studies in kidney fibrosis model in mice showed that vitamin D upregulates expression of multiple genes including endothelin-1 and VEGF, thereby limiting vascular remodelling and ischemia [106]. Majority of the anti-angiogenic information regarding vitamin D is derived from cancer studies and vitamin D was shown to exhibit anti-cancer effects by reducing vascularity [107]. Calcitriol negatively influences angiogenesis by inhibition of VEGF and induction of apoptosis in epithelial cells (melanoma) [108]. ECs derived from tumours from *Vdr^-/-^* mice showed that vitamin D decreased the proliferation of ECs and loss of VDR lead to an upregulation in the levels of angiogenic factors such as VEGF, HIF-1α, angiopoietin-1 and platelet-derived growth factor [109]. Previous studies also showed that 1,25(OH)_2_D_3_ lead to significant attenuation of retinal neovascularization in a mouse model of oxygen-induced ischemic retinopathy [110] and during retinal vascular development [111]. Calcitriol promotes pro-angiogenic molecules including VEGF and decreased angiogenin and HIF-1α expression in keratinocytes in a diabetic foot ulcer model and in ECs and keratinocytes in vitro [112]. Vitamin D intervention does not improve vascular regeneration in diet-induced obese male mice with peripheral ischemia [113]. The study shows that recovery of hind limb use in diet-induced obese mice following acute ischemic injury was improved in niacin-treated mice compared to vitamin D-treated mice suggesting limited benefit of vitamin D supplementation vascular regenerative processes. Well-designed preclinical studies using patient relevant models are necessary to identify the effect of vitamin D supplementation in PAD formation.

## 4. Vitamin D and the Genome

Physiologically, vitamin D is sourced from cutaneous radiation by ultraviolet B from the sun and dietary intake [7,114] (Figure 1). The initial, and most crucial step during vitamin D bio-activation is hepatic hydroxylation to 25(OH)D by the enzyme Cytochrome P450 Family 2-Subfamily R Member 1 (CYP27A1) [115,116,117]. Liver mitochondrial and microsomal 25-hydroxylases (25-OHases), encoded by CYP27A1, carry out the first hydroxylation of vitamin D to form 25(OH)D_3_. The main circulating form of vitamin D is 25(OH)D and assay for this metabolite are recommended for assessment of vitamin D status [6]. 25(OH)D circulates bound to DBP, accounting for both endogenous and exogenous sources. In addition to its transportation role, DBP also regulates the uptake of 25(OH)D by target cells [114]. Vitamin D bio-activation occurs mainly during renal hydroxylation of the circulating 25(OH)D to 1α,25-dihydroxyvitamin [1α,25(OH)_2_D] which is catalyzed by the enzyme CYP27B1 [1]. 

Recent advances in molecular techniques has facilitated elucidation of vitamin D genomic effects [118]. It has been shown that in order for vitamin D to mediate gene transcription, 1α,25-(OH)_2_D_3_ has to bind to the cytosolic/nuclear receptor, a member of the steroid/thyroid hormone receptor superfamily named VDR. The VDR is a member of the nuclear receptor family that regulates gene transcription by forming a hetero-dimer with RXR, which binds to vitamin D-response elements (VDRE) in the promoter regions of target genes [119] (Figure 2). This interaction is facilitated by a heterodimer formation with the RXR of the steroid receptor family members; which then communicates directly with DNA sequences located within regulatory regions of target genes (Figure 2). Formation of this complex facilitates exposure of unique binding sites on the surface of both the VDR and RXR resulting in recruitment of additional multi-protein co-regulatory complexes and this subsequently leads to either activation or inactivation of target genes. The gene product is transcribed in response to a positive or negative VDREs located in the promoters, enhancers or suppressors of the genes. The direct modulation of transcription by VDR ligands occurs if there is at least one activated VDR close to the transcription start site (TSS) of the specific primary target gene achieved through site directed (site-specific) binding to DNA (Figure 2). For example, the VDRE complex is located in Osteopontin gene (*SSP1*) and the *CYP24A1* gene. The active vitamin D when bound to VDR, regulates VDR expression through VDREs in its own enhancers, there by auto-regulating VDR.

The non-genomic pathways are activated via a putative membrane VDR and might be responsible for rapid effects of vitamin D. The VDR gene is located on chromosome 12q12-q14 in humans. Several frequent polymorphisms are also found in the VDR gene and were reported to be associated with a variety of phenotypes in many populations. However, to date the status of VDR gene polymorphisms was not assessed in PAD population. Interestingly, expression of VDRs and 1α-hydroxylase (1α-OHases) by the crucial cell types in the vasculature, including ECs, VSMCs and immune cells currently support the evidence that these cells have the ability to synthesize 1,25(OH)_2_D from circulating 25(OH)D [1,120]. Locally synthesised 1α,25-(OH)_2_D then mediates both autocrine and paracrine effects in these cells. The non-genomic effect of vitamin D also include regulation pathways of calcium and phosphate homeostasis, activating several kinases, which play a role in modulating various signalling pathways [121]. 

The binding of VDR, a nuclear transcription factor, to 1,25(OH)_2_D elicits physiological regulation of gene transcription [122] (Figure 2). The vitamin D endocrine system regulate approximately 3% of the human genome [123]. Genome wide analysis of human samples have enabled molecular profiling of vitamin D levels and the response to vitamin D3 supplementation [124]. A study of peripheral blood mononuclear cells (PBMNCs) and adipose tissue biopsies from elderly, pre-diabetic individuals, who were supplemented with vitamin D_3_ for more than 5 months (*n* = 71) showed the expression levels of multiple genes to be directly correlating with alteration in 25(OH)D_3_ levels. This shows that individuals can be classified as responders and non-responders based on molecular profiling to understand whether they will be benefitted prior to supplementation with vitamin D_3_ [124]. 

## 5. Vitamin D and the Epigenome

The epigenome is a crucial layer of information, which regulates and decides on the part of the genome which are accessible and thereby regulate which regions are active. The epigenome is the chromatin state of a cell, which encompasses the genomic location of DNA and histone modifications, transcription factor binding sites, and the 3-dimensional structure of the DNA. The DNA is packaged within the eukaryotic cell leading to a compact higher order stature forming the dense arrays of nucleosomes seen in heterochromatin regions (Figure 4). 

Despite being tightly packed, the chromatin is accessible to transcription factors due to the two major epigenetic mechanisms involved in the process; mainly Histone protein modifications and DNA methylation. Epigenetic signatures can thus be highly dynamic and lead to short-lived stages such as response of chromatin to stress signals [125]. Studies assessing the role of epigenetic mechanism in PAD formation is scarce [126,127]. Recently, we have shown that epigenetic silencing mechanisms results in AAA formation in human and in mice [128,129] indicating the importance of epigenetic mechanisms in disease formation. 

### 5.1. Histone Modifications

Histone proteins are bound to the DNA and post-translational modifications of the N-terminal tails of histone proteins allow the chromatin to relax, and genes to become activated. Histone modifications change in response to environmental stimuli [130]. Histones are major protein components of chromatin that undergo post-translational modifications, including acetylation of lysines, methylation of lysines and arginines, and phosphorylation of serine and threonine residues [131]. Histone acetylation generally correlates with transcriptional activation and is dependent on a dynamic interaction between histone acetyltransferases (HATs) and histone deacetylases (HDACs) [132].

Primary epigenetic effects of vitamin D are linked to promoting chromatin accessibility through histone modifications, mainly acetylation. Interaction with chromatin and its modifying enzymes is a central element in 1α,25(OH)_2_D_3_ signalling [133]. Vitamin D binds through the VDR to the VDRE, which results in transcriptional regulation through epigenetic mechanisms such as histone modification, chromatin remodelling and altered binding of RNA polymerase II [134,135] (Figure 4). For example, treatment of THP-1 cells with 1,25-D_3_ increased acetylation of histone protein (H3K27ac) at the promoter of several early VDR target genes [136]. Higher levels of histone acetylation were also found at specific regions in VDR and 1,25 (OH)_2_D_3_-induced binding of VDR to these specific sites was also associated with increased histone levels [137] (Figure 4). Furthermore, VDR reactivation induced by the HDAC inhibitor Trichostatin-A along with calcitrol was found to upregulate a group of repressed VDR gene targets controlling proliferation and induction [138,139]. It was also found that in the absence of ligand, VDR is able to bind to genomic DNA [140] and forms complexes with co-repressor proteins and HDACs [141]. Furthermore, the VDR-RXR dimer interacts with HATs to induce transcriptional activation [142].

Studies directly assessing histone modifications in PAD are meagre, however there is plenty of evidence in the literature which shows a strong effect of histone modification in mechanisms underlying PAD formation [126,143]. Most of the vitamin D responsive chromatic sites cluster at specific loci within the human genome and the most prominent of the region is on chromosome 6 which harbours the human Leukocyte antigen (HLA) region. This is significant as the HLA cluster is also a region important for vitamin D responsiveness to epigenome shows the strong immune regulation potential of vitamin D [144]. Studies shows that the primary activation step after 1α,25(OH)_2_D_3_-VDR complex is the transient opening or closing of the chromatin at specific enhancer and TSS with RNA polymerase II and other nuclear adaptor proteins, which determines the regulation (activation or suppression) of primary vitamin D target genes [145] (Figure 4). Nuclear Receptor Corepressor 1 (NCOR1) is a co-repressor that contains nuclear receptor interacting domains and interacts with various nuclear receptors, including VDR [146]. NCOR1 recruits histone deacetylase enzymes (HDACs) which deacetylase histone-tails and thereby condense the DNA structure and prevent gene-transcription. 1,25(OH)_2_D_3_ has been shown to increase NCOR1 gene expression [147] showing a direct epigenomic regulation role.

Two months of daily supplementation of vitamin D3 (10 μg/50 μg) resulted in significant regulation of 291 genes and weekly supplementation using a higher dose (500 μg) over 35–years resulted in significant changes in 99 genes in PBMNCs compared to placebo group [148]. The Phase I of the *VitDbol* vitamin D intervention trials [149] showed that a single bolus of vitamin D3 (2000 μg) results in a change in chromatin assembly in a selected genomic region [150]. This leads to a further proof of principle study during Phase II where one individual was exposed to vitamin D3 bolus for 3 consecutive days for every 28 days [149,150]. Time-point analysis of epigenome wide chromatin accessibility was assessed by applying a technique called formaldehyde assisted isolation of regulatory elements sequencing (*FAIRE-seq*). Results shows that under in vivo conditions, even a minor rise in serum levels of 1α,25(OH)_2_D_3_ is sufficient to result in significant changes at hundreds of sites within the epigenome of human leukocytes. The Phase II of *VitDbol* study conducted an RNA sequencing (*RNA-seq*) for the transcriptome-wide analysis of the effects of a vitamin D_3_ bolus on PBMCs from 5 individuals [151]. It was shown that 702 genes were being significantly modulated by vitamin D. This observation is supported by an elegant in vitro cell culture study that shows convincing evidence that the 1α,25(OH)_2_D_3_ significantly affect thousands of human genomic loci, promoting in major changes in the epigenome [152].

### 5.2. DNA Methylation

Alterations in DNA methylation lead to aberrant gene expression and disruptions of genomic integrity, which contribute to development and progression of PAD and vitamin D can regulate these processes. Recent research suggests the potential of vitamin D to modulate the gene expression through modulating the epigenetic marks [153]. Previous reports show that concentrations of 1,25(OH)_2_D_3_ in serum vary widely in individuals based on variations in diet, sun exposure, difference in age, adiposity and genetic and epigenetic variations [154,155,156]. Vitamin D regulates expression of genes in genome wide scale as evidenced by the activation of more than 200 genes in lymphoblasts stimulated by calcitriol [134]. Previous population-based cohort studies identifies different gene expression profiles in individuals with different concentrations of plasma 25(OH)D [157]. Primary human PBMNCs subjected to calcitriol (100–120 nM) exposure showed altered expression of known vitamin D responsive genes [158]. In healthy subjects global, age-related methylation of human rectal mucosa was influenced not only by gender, folate availability, and selenium, but also by vitamin D status [159].

Interestingly, a weak positive correlation was observed between vitamin D level with methylation of *LINE-1* (genomic long interspersed nuclear element-1), a mammalian autonomous retrotransposon, and increasing stability of this region [159]. A recent study associated severe vitamin D deficiency with methylation changes in leukocyte DNA, although the observed differences were relatively small [160]. This study suggested that subjects with vitamin D deficiency were more likely to show reduced synthesis and increased catabolism of active vitamin D. Whether this was the cause of the vitamin D deficiency or the consequence thereof is not clear and needs further studies. Furthermore, analysis of genome-scale methylation in leukocytes from vitamin D (plasma 25(OH)D) deficient adolescents in comparison to adolescents with sufficient circulating plasma 25(OH)D, showed several gene loci to be differentially methylated [160].

Vitamin D metabolism and DNA methylation may have a complex relationship [161]. Previous studies have shown that vitamin D signalling is controlled by epigenetic mechanisms, and stimulation using vitamin D influences global methylation markers [159]. Previous reports show that the major regulators of 1,25-D_3_ levels and “the vitamin D tool” genes signalling through *CYP2R1*, *CYP24A1*, *CYP27B1*, and *VDR*, are prone to epigenetic regulation [153]. For example, the tissue levels of the biologically active form of vitamin D (1,25(OH)_2_D_3_) is controlled though its biosynthesis by α–hydroxylase or CYP27B1. The 24-hydroxylase enzyme encoded by the *CYP27B1* gene is the main player in the catabolism of calciferol to less active vitamin D metabolites and previous studies shows that the *CYP27B1* gene expression is epigenetically modulated in various mammalian tissues [162,163]. *CYP2R1*, *CYP24A1* and *VDR* have CpG islands in their respective promoter regions and *CYP27B1* has a CpG island located in the gene body [153,164]. Hypermethylation of *VDR* promoter region generally influences VDR expression and function suggesting the crucial role of epigenetics in vitamin D signalling [165]. 

Altered DNA methylation has also been implicated in Vitamin D mediated regulation of expression of a number of genes [166,167,168]. For example, high dietary vitamin D intake was associated with lower methylation of the two Wnt antagonists *Dickkopf 1* (*DKK1*) and *WNT5A* [168]. The altered expression of the gene in the vitamin D downstream signalling pathway such as Wnt/β-catenin pathway will be crucial in AAA. Wnt inhibitor Sclerostin was recently reported to undergo DNA methylation in AAA [128]. Similarly, our recent report shows that in humans, serum angiotensin converting enzyme (ACE)-2 activity was negatively associated with AAA diagnosis [129]. Furthermore, the expression of *ACE-2* was lower in infra-renal biopsies of patients with AAA than organ donors due to dense methylation in the promoter region of *ACE-2* gene. The neural precursor cells expressed a ubiquitin ligases, the Developmentally down-regulated protein-4 (NEDD4) which function in the ubiquitin proteasome system of protein degradation. *Nedd4* deletion in mice results in deformed aortic structures with disarranged elastin fibres and vitamin D was shown to induce aorta calcification in these mice even at very low levels [169]. Furthermore, epigenetic analysis revealed that methylation levels of human *NEDD4* gene promoter were significantly increased in atherosclerosis patients suggesting that vitamin D supplementation might not be a possibility in all atherosclerotic diseases.

A 12-month interventional study in elderly people showed that intervention with a combination of vitamins (combination of vitamin D, B, and Calcium) differentially affected surrogate markers of epigenetic age, mainly CpG methylation of three age associated genes (Aspartoacylase, *ASPA*; cAMP-specific 3′,5′-cyclic phosphodiesterase 4C, *PDE4C* and integrin alpha-IIb, *ITGA2B*) compared with vitamin D + Calcium alone [170]. This study suggest that people can have higher biological age than chronological age, which might be due to accelerated aging as a result of hypovitaminosis. This is an important observation as accelerated senescence of cells is a crucial underlying pathological change occurring in age-associated diseases such as AAA. 

## 6. Future Direction

Vitamin D insufficiency has emerged as a public health issue in industrialised societies worldwide. Maintaining an optimal vitamin D serum level seems important not only for calcium homeostasis but also for minimising cardiovascular risk, blood pressure control, metabolic syndrome, prevalence of stroke, and PAD. The observational data support the strong link between epigenetic effects of vitamin D status and PAD formation. However, there is still a lack of clarity regarding the ideal threshold of vitamin D that is required to prevent PAD. However, consuming higher amounts of vitamin D may interfere with regulation of phosphate metabolism through Klotho gene product and fibroblast growth factor, which has detrimental consequences including aortic calcification [171]. It is therefore important that the ideal level of vitamin D is identified that can reduce the risk of PAD development while not having other deleterious effects associated with high doses.

There is strong evidence presented in this review, which suggests that the vitamin D system and epigenetic mechanisms are closely connected. The vitamin D system is shown to regulate various epigenetic mechanisms and also vitamin D system itself is regulated by those mechanisms. As it has been shown by multiple studies, the regulation of epigenetic mechanisms is finely tuned and any slight de-regulation can lead to pathological conditions due to its numerous roles in critical pathological systems. 

Considering the modern lifestyle of the increasing global aging population and indoor-oriented sedentary lifestyle, the epigenetic effects of vitamin D in promoting PAD is highly important for public health. Since the epigenome can be modified, consuming sufficient micronutrients including vitamin D may be an alternative method of prevention of any disease. Identifying the molecular mechanisms, the effect of vitamin D insufficiency and the link to PAD formation will unveil new therapeutic avenues and targets. The emerging field of nutri-genomics and nutri-epigenetics will pave way to identify and treat the PAD triggers early and also provide sustainable solutions to protect the arteries from developing occlusive disease. Even though there is evidence of vitamin D levels in PAD formation, the beneficial role of vitamin D supplementation still remains inconclusive and needs further research.

## Figures and Tables

**Figure 1 ijms-20-04907-f001:**
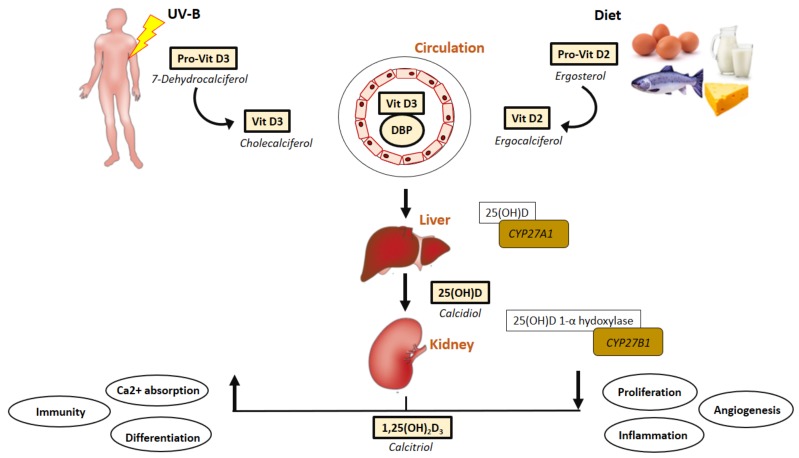
Schematic illustration vitamin D synthesis pathway and signalling mechanisms relevant to PAD formation. The main forms of vitamin D in nature are vitamin D3 (cholecalciferol) that is synthesized in the skin of animals and humans in response to sunlight and obtained through diet. The vitamin D3 travels in the circulation bound to DBP. Vitamin D must undergo several hydroxylation steps to become an active metabolite. The synthetic pathway involves 25- and 1-alpha-hydroxylation of vitamin D3 and D2, in the liver and kidney, respectively. The first hydroxylation occurs in the liver resulting in the formation of 25(OH)D3 or calcidiol and the second hydroxylation occurs mainly within the kidneys and intestinal epithelial cells and immune cells and generates the most biologically active hormonal form of vitamin D: 1, 25(OH)_2_D, or calcitriol. The biologically active form of vitamin D3 is involved in the regulation of numerous cell cycle regulatory mechanisms protecting the vasculature from pathological conditions. Abbreviations: Ca^2+^, calcium; DBP, vitamin D binding protein; PAD, peripheral arterial disease; Vit D3, vitamin D3_._

**Figure 2 ijms-20-04907-f002:**
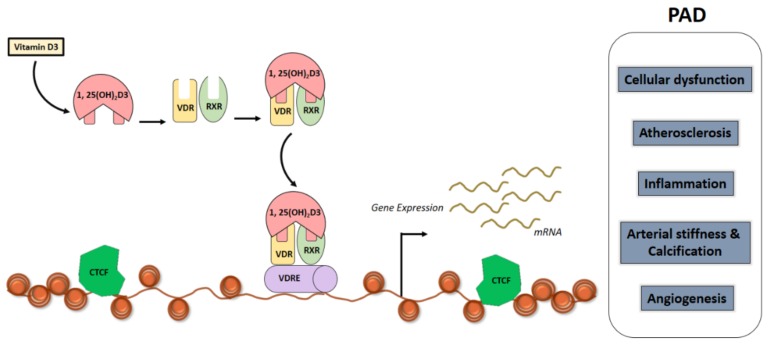
Schematic illustration vitamin D canonical signalling pathway and its role on mechanisms relevant to PAD formation. The biologically active form of vitamin D3 is 1,25(OH)_2_D_3_, is a ligand for a nuclear transcription factor VDR. VDR is present in most cell types and once the VDR is bound with 1,25(OH)_2_D_3_, then it translocates to the nucleus. VDR has a binding site that recognises the target sequence in the genome. VDR binds to genomic DNA after it dimerizes with RXR. The genomic region that binds to VDR and 1,25(OH)_2_D_3_ is restricted by CTCF protein, which defines the TAD. CTCF is a transcription factor which is often found near the TSS on the chromatin. For 1,25(OH)_2_D_3_ to initiate transcription, the target gene should have its TSS and at least one VDR binding site at a location within the same chromatin. The vitamin D target genes that fall within the TAD region of a chromatin will undergo gene expression. Genes containing a specific promoter region known as VDRE will allow the binding of VDR-RXR dimers to the VDRE region and recruit transcriptional machinery including transcription factors and RNA polymerase II. Abbreviations: CTCF, CCCTC-binding factor; PAD, peripheral arterial disease; RXR, Retinoid X Receptor; TAD, Topologically Associated Domain; TSS, Transcription Start Site; VDR, Vitamin D Receptor; VDRE, Vitamin D Response Elements.

**Figure 3 ijms-20-04907-f003:**
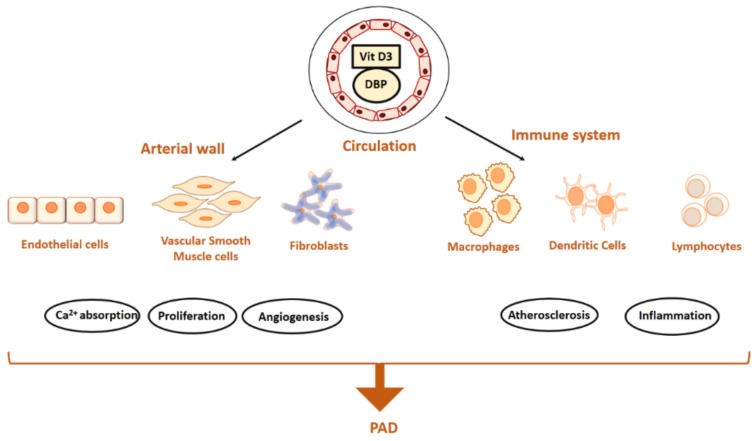
Schematic illustration summarising the effect of vitamin D on various resident cells and its role in modulating the molecular mechanism involved in PAD formation. Vitamin D has been proposed to be involved in the regulation of many cell types involved in PAD formation, mainly the residential vascular cells such as endothelial cells, vascular smooth muscle cells, firboblasts and also infiltrating and/or circulating immune cells. Vitamin D has shown to be potent growth modulator, protector of endothelial function and inflammatory response mediator on the resident vascular cells playing a crucial role in atherogenesis. Optimal vitamin D status is crucial for the functioning of the resident cells and thus it plays a major protective effect in vascular wall promoting matrix integrity. *Abbreviations*: Ca^2+^, calcium; DBP, vitamin D binding protein; PAD, peripheral arterial disease; Vit. D3, vitamin D_3_.

**Figure 4 ijms-20-04907-f004:**
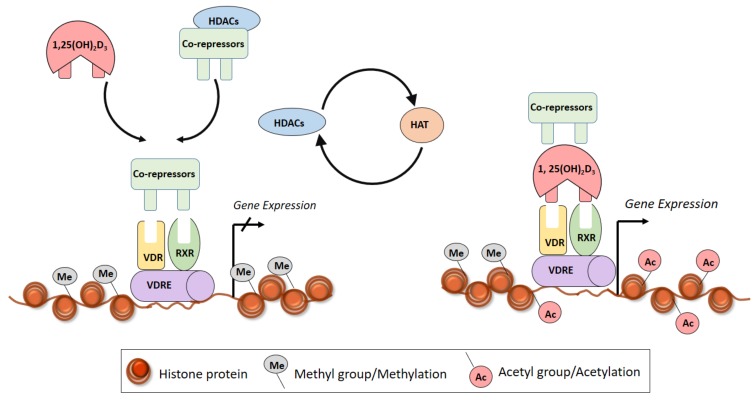
Schematic illustration of histone modification mechanisms mediated by vitamin D3 and the link to basal transcription machinery. The DNA is packaged within the eukaryotic cell leading to a compact higher order stature forming the dense arrays of nucleosomes seen in heterochromatin regions. Since the chromatin is tightly packed, the chromatin is in-accessible to transcription factors. The direct protein-protein interaction with co-activators which have HAT activity leads to local chromatin opening. Two major epigenetic mechanisms facilitates relaxation of chromatin structure and promotes gene expression; mainly DNA methylation and Histone protein modifications. (**a**) DNA methylation in the promoter region of VDR gene results in the inactivation and lack of VDR gene expression. (**b**) When HDAC-corepressor complexes are bound to the heterochromatin region, the transcriptional machinery is repressed resulting in downstream gene silencing. (**c**) When VDR-RXR complex is bound to HATs, HAT transfers acetyl group to histones leading to histone acetylation and chromatin remodelling. The chromatin machinery is then relaxed resulting in binding of TFs and activation of expression of downstream genes. Abbreviations: HAT, Histone Acetyl Transferase; HDAC, Histone deacetylase; RXR, Retinoid X Receptor; VDR, Vitamin D Receptor; VDRE, Vitamin D Response Elements.

**Table 1 ijms-20-04907-t001:** Cross-sectional studies investigating vitamin D circulating levels in peripheral arterial occlusive disease (PAOD) patients.

Studies	Type of Study	Country	Main Findings	Refs
Rapson IR, et al. 2017	Prospective	USA, ARIC study cohort	Deficiency of 25(OH)D was associated with increased risk of PAOD in black and white participants	[23]
Liew JY, et al. 2015	Cross-sectional (case-control)	Australia	No significant difference 25(OH)D levels was detected between PAOD patients (only IC) and control	[32]
Veronese N, et al. 2015	Prospective	Italy, community dwelling men	Baseline hypovitaminosis D (<24 nmol/L) did not predict the onset of PAOD over a 4.4-year follow-up in elderly people	[33]
Amer M, et al. 2014	Retrospective	USA	Elevated serum 25(OH)D concentration was associated with significant increase in ABPI in asymptomatic adults without PAOD	[24]
Stricker H, et al. 2012	Double-blind, placebo-controlled	Caucasian, Switzerland	PAOD patients had low 25(OH)D levels (<30 ng/mL)	[25]
McDermott MM, et al. 2012	Cross-sectional (case-control)	USA	No significant difference 25(OH)D levels was detected between PAOD patients (only IC) and control	[34]
Gaddipati VC, et al. 2011	Cross-sectional	USA	Deficiency of 25(OH)D (<20 ng/mL) was associated with an increased amputation risk in veterans with PAOD	[26]
Zagura M, et al. 2011	Cross-sectional (case-control)	Estonia	PAOD patients had lower 25(OH)D compared to controls	[27]
Melamed ML, et al. 2008	Cross-sectional (case-control)	USA, nationally representative adults >20years of age	Lower serum 25(OH)D levels are associated with a higher prevalence of PAOD	[28]
Reis JP, et al. 2008	Cross-sectional	USA	After adjustment for 25(OH)D levels, odds for PAOD were reduced from 2.11 (95% CI: 1.55, 2.87) to 1.33 (95% CI: 0.84, 2.10) in black compared with white participants	[29]
Fahrleitner-Pammer, et al. 2005	Cross-sectional (case-control)	Austria	Patients with CLI symptoms had lower 25(OH)D compared to both IC and controls	[30]
Fahrleitner A, et al. 2002	Cross-sectional (case-control)	Austria	Patients with CLI symptoms had lower 25(OH)D compared to both IC and controls	[31]

Abbreviations: ABPI, ankle brachial plexus index; ARIC, Atherosclerosis Risk In Communities; CLI, critical limb ischemia; IC, intermittent claudication; PAOD, peripheral arterial occlusive disease.

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
