# Peer review of "Vitamin D as A Protector of Arterial Health: Potential Role in Peripheral Arterial Disease Formation"

_ijms, 2019, doi:10.3390/ijms20194907_

Round 1

Reviewer 1 Report

The review paper by Smriti Murali provides a literature update on the role of Vitamin D in peripheral arterial disease. It is clear, well-written and informative.

Here some comments:

We agree that sunlight is an important requirement for vitamin D synthesis in the skin, but can it be considered the main source as mentioned by author in two paragraphs? Please consider reviewing. Author highlighted that the Vitamin D levels that define hypovitaminosis differ significantly between studies. Does this relate to geography? If there is lack of standardization, can the authors suggest his views on this? Similarly to above, author describes in detail, many contradictory results for Vitamin D effects. It is important to clarify shortcomings or limitations of key studies, or any other reasons to explain such discrepancy. An additional figure or table to summarize the molecular mechanisms of Vitamin D in calcification, inflammation, and angiogenesis would be useful to help the reader navigate through the complexity of such research. There are two different font within the manuscript. Some typrographical errors, for example line 229, is it RAS or RAAS?

Reviewer 2 Report

This is an interesting review article aiming to update the readers on the status of research involving Vitamin D as a protector of vascular health with a specific focus on the role of Vitamin D in peripheral artery disease formation.

This is an important subject and needs our attention since we are facing an increase in the prevalence in our aging society. Still no effective preventive or treating measures are implement in the clinical practice. In general, the manuscript is very well and comprehensively written.

Of note, the author provides the reader an open discussion referring the conflict of information that is still available from study on this research subject and addressing a diversity of aspects from in vitro to clinical evidences, always including available mechanistic studies.

Nevertheless, the review as some defaults namely some incomplete sentences, typos and incorrect references that should be corrected and will be described under detailed comments.

Detailed comments:

Based on the comments made before, the authors should correct the following according to the suggestions below.

In page 2, line 53, a typo error needs correction

In page 3 legend figure 1 line 81-82, revise the abbreviations description

In page 3 line 96, revise sentence

In page 4 line 130, revise sentence

In page 5 line 149 please use the full description of AAA (Abdominal aortic aneurysms) the first time in this  2.2 section

In pag2 6 legend text revise the sentence in  line 219-221

In page 6 line 229, correct the abbreviations used (RAAS instead of RAS)

In section 3.1 it is not clear the inclusion of Figure 2 in the text as a reference for the reader in this particular sentence.

 Also in this section, line 253, correct the  eNOS abbreviation according with the abbreviation list included in the manuscript to homogenise. Also please correct and homogenise the use of Vdr -/- abbreviation when referring to the knock out mice and to the VDR. Again referred in line 373.

In line 273 section 3.2, when referring the results from reference 80, the author should clarify that the population used in this study was diabetic and discuss this fact.

 In line 328, it seems that the reference to the same study (reference 80) is misplaced.

Also in line 338, reference 93 does not include direct results on arterial stiffness and this should be addressed to comply with the parameters evaluated in this study, which were Brachial Artery FMD and Endothelium Independent Dilation.

In section 6 sentence in line 587-588, needs to be corrected.
